# Application of Single Cell Type-Derived Spheroids Generated by Using a Hanging Drop Culture Technique in Various *In Vitro* Disease Models: A Narrow Review

**DOI:** 10.3390/cells13181549

**Published:** 2024-09-14

**Authors:** Hiroshi Ohguro, Megumi Watanabe, Tatsuya Sato, Nami Nishikiori, Araya Umetsu, Megumi Higashide, Toshiyuki Yano, Hiromu Suzuki, Akihiro Miyazaki, Kohichi Takada, Hisashi Uhara, Masato Furuhashi, Fumihito Hikage

**Affiliations:** 1Departments of Ophthalmology, Sapporo Medical University, S1W17, Chuo-ku, Sapporo 060-8556, Japan; watanabe@sapmed.ac.jp (M.W.); nami076@yahoo.co.jp (N.N.); araya.umetsu@sapmed.ac.jp (A.U.); megumi.h@sapmed.ac.jp (M.H.); 2Departments of Cardiovascular, Renal and Metabolic Medicine, Sapporo Medical University, S1W17, Chuo-ku, Sapporo 060-8556, Japan; satatsu.bear@gmail.com (T.S.); oltomwaits55@gmail.com (T.Y.); furuhasi@sapmed.ac.jp (M.F.); 3Departments of Cellular Physiology and Signal Transduction, Sapporo Medical University, S1W17, Chuo-ku, Sapporo 060-8556, Japan; 4Departments of Molecular Biology, Sapporo Medical University, S1W17, Chuo-ku, Sapporo 060-8556, Japan; hsuzuki@sapmed.ac.jp; 5Departments of Oral Surgery, Sapporo Medical University, S1W17, Chuo-ku, Sapporo 060-8556, Japan; amiyazak@sapmed.ac.jp; 6Departments of Medical Oncology, Sapporo Medical University, S1W17, Chuo-ku, Sapporo 060-8556, Japan; ktakada@sapmed.ac.jp; 7Departments of Dermatology, Sapporo Medical University, S1W17, Chuo-ku, Sapporo 060-8556, Japan; uhara3@yahoo.co.jp

**Keywords:** 3D spheroid culture, hanging drop culture, *in vitro* model, gravity force, buoyant force

## Abstract

Cell culture methods are indispensable strategies for studies in biological sciences and for drug discovery and testing. Most cell cultures have been developed using two-dimensional (2D) culture methods, but three-dimensional (3D) culture techniques enable the establishment of *in vitro* models that replicate various pathogenic conditions and they provide valuable insights into the pathophysiology of various diseases as well as more precise results in tests for drug efficacy. However, one difficulty in the use of 3D cultures is selection of the appropriate 3D cell culture technique for the study purpose among the various techniques ranging from the simplest single cell type-derived spheroid culture to the more sophisticated organoid cultures. In the simplest single cell type-derived spheroid cultures, there are also various scaffold-assisted methods such as hydrogel-assisted cultures, biofilm-assisted cultures, particle-assisted cultures, and magnet particle-assisted cultures, as well as non-assisted methods, such as static suspension cultures, floating cultures, and hanging drop cultures. Since each method can be differently influenced by various factors such as gravity force, buoyant force, centrifugal force, and magnetic force, in addition to non-physiological scaffolds, each method has its own advantages and disadvantages, and the methods have different suitable applications. We have been focusing on the use of a hanging drop culture method for modeling various non-cancerous and cancerous diseases because this technique is affected only by gravity force and buoyant force and is thus the simplest method among the various single cell type-derived spheroid culture methods. We have found that the biological natures of spheroids generated even by the simplest method of hanging drop cultures are completely different from those of 2D cultured cells. In this review, we focus on the biological aspects of single cell type-derived spheroid culture and its applications in *in vitro* models for various diseases.

## 1. Introduction

In addition to conventional two-dimensional (2D) culture methods, much attention has recently been given to three-dimensional (3D) cell cultures that can replicate microenvironments of cells under physiological and pathological conditions, and applications of 3D culture methods in various biological research fields have been rapidly increasing [1]. There are various 3D cell culture methods ranging from the simplest single cell type-derived spheroid cultures to more sophisticated methods including spheroid co-cultures, spheroid on-chip cultures, 3D organoid cultures, tissue-like organoid cultures, and 3D organoid on-chip cultures [2,3,4]. Furthermore, for the simplest method, the single cell type-derived spheroid method, there are various methods using biomaterial scaffolds for assisting the formation of spheroids including hydrogels, biofilms, particles, and magnet particles and methods that do not require scaffolds including static suspension cultures, floating cultures, and hanging drop cultures [3]. Recently, we have been carrying out studies on the pathophysiology of various non-cancerous and cancerous tissues using single cell type-derived spheroids generated by using the hanging drop culture method [5,6,7,8,9]. Based on our experience, we mainly focus on single cell type-derived spheroid methods in this review and discuss the advantages and disadvantages as well as their contributions to elucidation of the pathophysiological aspects of various non-cancerous and cancerous diseases.

### 1.1. Various Methods for Single-Cell 3D Spheroid Generation

#### 1.1.1. Scaffold-Assisted Methods

##### Hydrogel-Assisted Cultures

Hydrogels are crosslinked polymer chains with 3D spatial structures that can contain a large amount of water due to their hydrophilic groups such as -NH2, -COOH, -OH, -CONH_2_, -CONH, and -SO_3_H [10]. In practical use, these scaffolds are placed in a well [11], in a microfluidic channel [11], or on a micropillar [12]. An ad-vantage of this culture method is that the physical properties, including the size and stiffness, of spheroids can be optimized by selecting biomaterials such as alginate [13], fibrin [14], collagen [15], hyaluronic acid [16], and ECM-based hydrogels, such as Mat-rigel^®^ [17,18,19] and Cultrex^®^ [20], and their appropriate concentrations for fabrication of 3D spatial scaffolds [21].

##### Biofilm EPS-Assisted Cultures

Biofilm extracellular polymeric substances (Biofilm EPSs), which consist of a mix-ture of polysaccharides, extracellular DNA, and proteins [22], are used for spheroid generation [23]. It was shown that the size of spheroids can be controlled by the thick-ness of the film and composition of the EPS [24,25].

##### Gelatin Microparticle-Assisted Cultures

In this culture method, cells are cultured with gelatin microparticles containing biological factors on a cell repellent-coated culture plate [26,27]. By using desired bio-logical factors in the particles, the biological nature of the spheroids can be spatially controlled [28,29].

##### Magnetic Particle-Assisted Cultures

In this method, cells are mixed with magnetic particles and cultured on a mag-net-equipped cell culture plate. Because of the magnetic force, cells aggregate with floating magnetic particles in the culture medium [30]. However, it has been suggested that artificially manipulated gravity may induce changes in the cellular biological na-ture that result in apoptosis [31,32].

#### 1.1.2. Biomaterial Non-Assisted Methods

##### Static Suspension Cultures

In this culture method, cells are concentrated to form a spheroid on culture plates with non-adhesive surfaces, such as ultra-low attachment (ULA) plates with U-shaped bottomed wells or surfaces coated with hydrophilic substances such as agar or poly Hema, to minimize cell–substrate adhesion so as to allow cells to remain suspended. Subsequent centrifugation is required to support the aggregation of cells and the formation of a single spheroid per well [33]. This culture method has been shown to generate reproducible spheroids with uniform sizes and shapes that are suitable for high-throughput drug screening [34]. To overcome the problem of limited generation of spheroids, a new 96-well plate microcavity array-based platform, which can generate spheroids in the microcavities, has recently been developed for drug testing with higher parallelization capacity than that of previous devices [35,36].

##### Floating Cultures

In this culture method, cells are cultured in a spinner or rotation chamber in which the cell suspension is continuously mixed by stirring to avoid cell adhesion to the bottom of the chamber [37]. According to the difference in intercellular affinity, the conditions of the stirring forces should be adjusted for proper spheroid formation [38].

##### Hanging Drop Cultures

This technique allows cells to spontaneously aggregate for the formation of a spheroid in a culture droplet. It is possible to control the spheroid size by adjusting the volume of the droplet or the concentration of cells in the suspension [39].

#### 1.1.3. Comparison of the Various Influencing Factors among the 3D Spheroid Culture Methods

Table 1 shows a comparison of the (1) sites of spheroid generation, (2) effects of gravity force, (3) effects of buoyant force, and (4) contributions of additive factors among the above-described culture methods. In the magnetic particle-assisted cultures, floating cultures, and hanging drop cultures, spheroids are not generated on the ground and are thus more influenced by gravity force and buoyant force compared to the other culture methods. In addition, except for static suspension cultures, other methods are influenced by additive factors including some scaffolds, magnet force, centrifugal force, or methyl cellulose as a spheroid stabilizer. Therefore, collectively, hanging drop culture is the simplest method for generation of spheroids as it is only influenced by gravity force, buoyant force, and methyl cellulose. In fact, a previous study showed that the hanging drop culture method generated circular spheroids that had a narrow distribution of sizes with variation coefficients of 10% to 15%, which are smaller than the variation coefficients of 40% to 60% in spheroid preparation using non-adherent surface culture methods [40]. Furthermore, by using a 384-hanging drop array culture plate, very efficient and reproducible formation of spheroids with controlled sizes is possible [41].

## 2. Preparation and Characterization of 3D Spheroids Formed by Using a 384-Hanging Drop Array Culture Plate

### 2.1. Representative Protocol of Preparation (Figure 1)

Step 1. Cells to be used for spheroid generation are prepared by conventional 2D planar culture in 100 mm or 150 mm dishes until the cells reach approximately 90% confluence at 37 °C using a CO_2_ cell incubator, as described elsewhere.Step 2. The cell pellet is collected from the 2D cell culture plate using 0.25% Tryp sin/EDTA and centrifugation after washing with phosphate-buffered saline (PBS) on a clean bench.Step 3. The cells are resuspended in the 2D culture medium supplemented with methylcellulose (Methocel A4M, Sigma-Aldrich Co., St. Louis, MO, USA) to facilitate stable morphology at a concentration of 20,000 cells in 28 μL of the culture medium on a clean bench.Step 4. A 28 μL aliquot of the cell suspension is placed in each well of a hanging drop culture plate (# HDP1385, Sigma-Aldrich Co., St. Louis, MO, USA) on a clean bench, and then the culture plate is placed in a CO_2_ cell incubator.Step 5. Half of the medium (14 μL) in each well is replaced daily by 14 μL of a fresh culture medium on a clean bench, and then the culture plate is placed in a CO_2_ cell incubator.

An important point is that careful attention should be paid to performing the procedures during steps 4 and 5 very gently to prevent the culture medium containing spheroids from dropping off. Another important point is that spheroid culture should be carried out during a period of approximately 4 to 12 days to accomplish its maturation, which is checked by a phase contrast microscopic examination. A longer culture period may lead to apoptosis of cells located near the core of the spheroid.

**Figure 1 cells-13-01549-f001:**
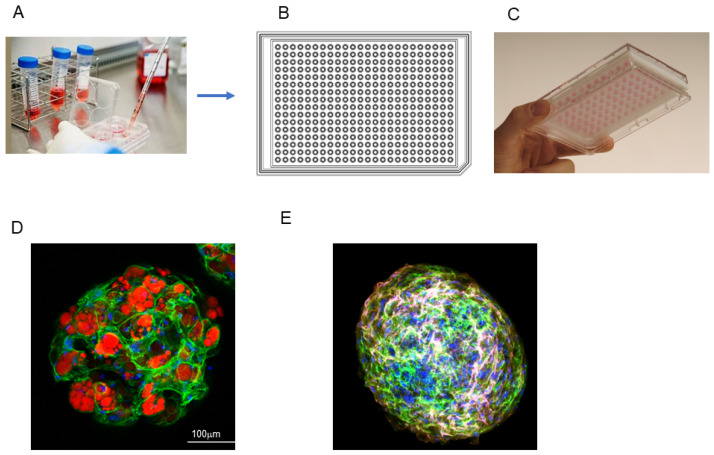
Method for preparation of 3D spheroids by a hanging drop culture plate. Cells obtained by conventional 2D planar culture (**A**) formed spheroid cultures by using a 384-hanging drop array culture plate ((**B**): design drawing). As shown in the upward view (**C**), a drop of culture medium containing cells hangs down from each well. Representative images of a spheroid of mouse orbital fibroblasts with adipogenesis stained by BODIPY (red), phalloidin (green), and DAPI (blue) (**D**) and a spheroid of human orbital fibroblasts with adipogenesis stained by anti-hyaluronic acid (pink), anti-COL6 (green), and DAPI (blue) (**E**).

### 2.2. Analyses of the Physical Properties, Including Size and Stiffness, and the Morphology of 3D Spheroids

#### 2.2.1. Measurement of the Sizes of 3D Spheroids

For measurement of the sizes of 3D spheroids, a phase contrast (PC) microscopy image is taken and then the cross-sectional area of the PC image is drawn and the area is calculated by using Image-J software version 1.51n (National Institutes of Health, Bethesda, MD, USA).

#### 2.2.2. Measurement of the Stiffness of 3D Spheroids

Measurement of the stiffness of a single living spheroid is carried out by using a MicroSquisher (CellScale, Waterloo, ON, Canada) consisting of a microscale cantilever-type pressure sensor with 3 mm × 3 mm basal and compression plates in a measurement chamber and a microscope with a camera outside the chamber. After the measurement chamber is filled with fresh PBS at 37 °C, a living single spheroid is transferred onto the basal plate and then gradually compressed by the compression plate with a compression bar until the vertical length of the spheroid becomes half of the length during a period of 20 s (Figure 2). The maximum force required (μN) is measured. The index for the stiffness of a spheroid is determined by maximum force/half of the length (μN/μm).

#### 2.2.3. Analysis of Morphology by Scanning Electron Microscopy and Immunocytochemistry

To study the morphology of a 3D spheroid in detail, the 3D spheroid is fixed with 2.5% glutaraldehyde for evaluation by scanning electron microscopy and is fixed with 4% paraformaldehyde for evaluation by immunocytochemistry and is processed as described elsewhere.

### 2.3. Representative Process of 3D Spheroid Maturation

As for the maturation process of a spheroid using a hanging drop culture method, our group showed that 3T3-L1 preadipocytes started to coalesce at 1 h after cells were added into wells of the culture plate and had formed premature spheroids by 6 h. The formation of spheroids became more apparent with the formation of smaller, round-shaped, and stiffened mature spheroids from day 1 through day 7 of culture [42]. Similar to 3T3-L1 spheroids, as shown in PC images (Figure 3), after placing 20,000 H9c2 cardiomyoblasts in 28 μL of culture medium into each well of a spheroid drop culture plate, the cells start to aggregate and then gradually form an immature spheroid within 7.5 h. On day 1, a matured spheroid formed.

### 2.4. Estimation of the Total Number of Cells in a Single Spheroid

As shown in trypsin digestion of 2D H9c2 cells and a H9c2 spheroid (Figure 4), once a spheroid has formed, the time needed to break the spheroid into single cells is much longer than that in the case of 2D cultured cells. Most of the cells in a spheroid are arranged concentrically from the center to the surface, as shown in the DAPI nuclei staining image (Figure 5). Therefore, to estimate the total number of cells in a spheroid, the tentative diameter of a spheroid is measured by its cross-section in a Phalloidin image and the diameter of a single cell in the spheroid is estimated by the distance between two adjacent nuclei stained by DAPI. Using these diameters, the number of cells in a spheroid is calculated by dividing the volume of the overall spheroid by the volume of a cell.

### 2.5. Cellular Metabolic Analysis of 3D Spheroid

The use of Seahorse extracellular flux bioanalyzers enables real-time measurements of oxygen consumption rate (OCR) as a function of mitochondrial respiration and extracellular acidification rate (ECAR) as a glycolysis function occurring mainly in 2D cultured cells [43]. In addition, several past studies have applied OCR and ECAR measurements in a 3D culture system using single-well assays [44,45,46,47]. This system was later optimized to measure OCR and ECAR in specimens of multiple spheroids using 96-multi-well plates [48]. This system has been used mainly in cancer research [48,49,50,51]. Our group has also performed measurements of OCR and ECAR of single cell type-derived spheroids using cancerous cells, including A549 cells [52], Mia PaCa2 cells [53], malignant melanoma (MM) cells [7,54], oral squamous cell carcinoma (OSCC) cells [55], and cancer-associated fibroblasts (CAFs) [9], and non-cancerous cells, including 3T3-L1 cells [56], human trabecular meshwork (HTM) cells [57,58], and human conjunctival fibroblasts (HconFs) [59]. Based on these experiences, we confirmed that the essential biological functions of mitochondrial respiration and glycolysis of spheroids are indeed substantially different from those of 2D cultured cells even though the same cells were used with only a difference in their culture system: 2D vs. 3D.

## 3. Application of 3D Spheroids for Various Fields in Biological Science

### 3.1. Study of Adipogenesis Using 3T3-L1-1 Cells (Figure 6)

In the field of adipogenesis-related research, mouse preadipocyte 3T3-L1 cells are most frequently used [60,61]. Although conventional 2D planar cell culture methods have been used in most of the studies in this field, for a better understanding of the in vivo nature of adipogenesis of adipose tissues in spatial environments in vivo, various 3D cell culture methods including polymeric scaffold-assisted cultures [62,63,64,65,66,67], magnetic particle-assisted cultures [68,69], and static suspension cultures using an elastin-like polypeptide-polyethyleneimine (ELP-PEI)-coated surface [70,71], in addition to hanging drop cultures using a 384-hanging drop array culture plate, have been used [42,72,73,74,75,76]. Among these different 3T3-L1 spheroid culture methods, our hanging drop culture is much faster and reproducibly produces round-shaped 3T3-L1 spheroids, although most of the studies in which spheroid culture methods were used have shown that biological aspects, such as the adipogenesis efficacy of 3T3-L1 spheroids, are significantly different from those of 2D 3T3-L1 cells. Recently, our group [6] and another study group [62] have shown spontaneous adipogenesis before adipogenic differentiation in 3T3-L1 spheroids but not in 2D cultured 3T3-L1 cells. In fact, proteomics [51] and RNA sequencing analysis [6,56] have revealed significant changes in various adipogenesis-related factors, inflammatory cytokines, growth factors, transcription factors, mitochondrial and glycolytic-related factors, and other factors during spheroid generation [44]. Furthermore, ingenuity pathway analysis (IPA) suggested that STAT3 is the master upstream regulator for inducing 3T3-L1 spheroid generation by comparing 2D and 3D cultured 3T3-L1 preadipocytes [6]. Since STAT3 has been shown to be involved in some gravity-induced biological effects Since STAT3 has been shown to be involved in some gravity-induced biological effects [77,78,79], this suggests that gravity force pivotally affects spheroid generation. In addition, our recent RNA sequencing analysis before and after adipogenic differentiation showed that genes related to adipogenesis and lipid metabolism were altered in 3T3L-spheroids compared to those in 2D cultured 3T3-L1 cells, suggesting that a spheroid environment may be more suitable for adipogenesis of 3T3-L1 cells [56].

**Figure 6 cells-13-01549-f006:**
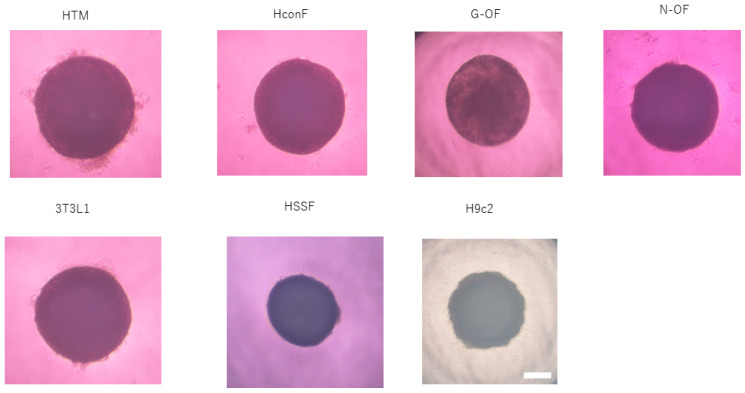
Representative images of spheroids obtained from various non-cancerous cells. HTM: human trabecular meshwork cell, HconF: human conjunctival fibroblast, G-OF: Graves-related human orbital fibroblast, N-OF: non-Graves-related human orbital fibroblast, 3T3-L1: 3T3-L1 mouse preadipocyte, HSSF: human scleral stromal fibroblast, H9c2: H9c2 rat cardiomyoblast. Scale bar: 100 μm.

### 3.2. Study for Cardiology Using H9c2 Cells (Figure 6)

The metabolic aspects of cardiomyocytes are known to be different from those of other myocytes as well as non-myocytes [80]. In fact, it has been shown that matured cardiomyocytes require most of their energy from fatty acid oxidation rather than glycolysis [81,82] and that cellular metabolic functions are deteriorated by aging and/or cardiac functional defects [83,84]. For instance, insufficient oxygen (O_2_) perfusion into cardiac muscle tissues in the case of myocardial infarction or other etiology induces heart failure [85], suggesting that oxygen (O_2_) supply is an indispensable key factor for maintenance of the physiological functions of the heart. For studying the pathophysiology of the heart and for drug screening of the heart, much attention has been paid to *in vitro* 3D cell culture models using cardiomyocytes [86,87] for replicating in vivo spatial environments in addition to the standardized two-dimensional (2D) cell culture method. 

H9c2 cardiomyoblasts originating from embryonic rat ventricular tissue are the most commonly used cells for cardiomyocyte research, and these cells have been applied to studies using different 3D culture methods including various scaffold-assisted methods [88,89,90,91,92,93,94,95,96,97,98,99], magnetic particles-assisted method [100] and floating culture using a rotating flask [101] in addition to hanging drop culture [102]. In those studies, *in vitro* 3D culture models were used cardiotoxicity screening and for investigation of drug-induced effects [89,100,101,102], apoptosis and cell survival [92,94], physical properties [91,102], cardiac homeostasis [88], myogenesis, differentiation and proliferation [93,96,97,99], cardiac disease modeling [98,102] and regeneration [95,96]. Great advantages of the 3D culture models compared to conventional 2D planar cell culture were shown in most of those studies. In fact, in our recent study, spontaneous expression of gap junction molecules and hypoxia-inducible factors (HIFs) were observed in H9c2 spheroids, but not in 2D H9c2 cells, and mitochondrial and glycolytic metabolic functions in H9c2 cells were significantly different in 2D and 3D culture conditions, suggesting that H9c2 spheroids indeed replicate essential physiological elements of cell-to-cell interaction in cardiomyocytes and O_2_ gradients in cardiac wall tissue [102]. Collectively, the results of studies indicated that an *in vitro* model of H9c2 spheroids is a useful *in vitro* model for understanding cardiac pathophysiology.

## 4. Studies for Ocular Pathophysiology

Three-dimensional cell cultures have been carried out using various parts of ocular tissues, and remarkable progress has recently been made in 3D organoid culture, but not spheroid culture, using human induced pluripotent stem cells (hiPSCs) for future retinal regenerative medicine [103]. Alternatively, spheroid culture methods have contributed to a better understanding of pathophysiology as well as various drug-induced effects in the ophthalmology research field as described in the following.

### 4.1. Ocular Surface and Sclera

*In vitro* spheroid models have been established using lacrimal gland (LG) cells and conjunctival cells by static suspension culture [104,105] and magnetic particle-assisted culture [106] to study tear-film dynamics and dry eye pathogenesis. Alternatively, our group established an *in vitro* subconjunctival fibrosis model using human conjunctival fibroblasts (HconF, Figure 6) by a hanging drop culture method to study the effects of various drugs including transforming growth factor-β (TGF-β isoforms [107], fibroblast growth factor-2 (FGF-2) [108], Rho-associated coiled-coil forming kinase (ROCK) inhibitors [109], all-trans retinoic acid (ATRA) [110], prostaglandin EP2 and FP2 agonists [111] and rosiglitazone [109]. For corneal disease modeling, spheroids were generated from bovine corneal endothelial cells by using a hydrogel-assisted method [112,113]. In addition, our group generated spheroids using human corneal stromal fibroblasts by a hanging drop culture method and we used the spheroids to study drug-induced effects of an EP2 agonist [114] and ROCK inhibitors [115]. We also established *in vitro* myopia models using spheroids that were generated from human scleral stroma fibroblasts (Figure 6) [116]. 

### 4.2. Intraocular Segments

To mimic *in vivo* microenvironments of the ocular lens, a particle-assisted method was used for generation of spheroids using lens epithelial cells [117]. To study aqueous humor (AH) dynamics related to glaucoma pathogenesis, various *in vitro* 3D models that replicate the trabecular meshwork (TM), which is the critical segment to produce AH outflow resistance during the AH conventional outflow route, were used (Figure 6) [118,119]. Scaffold-assisted 3D culture of TGF-β2 or dexamethasone (DEX)-treated human TM (HTM) cells was used in most of those studies, and those studies showed overexpressed ECM, defective phagocytosis of HTM cells resulting in increased resistance to AH outflow [120,121,122] and increase in the production of reactive oxidative species (ROS) [123]. Our group also obtained similar results using a hanging drop culture method without any scaffolds, and we successfully established *in vitro* primary open glaucoma (POAG) TM and steroid-induced glaucoma (SG) TM in the presence of TGF-β2 and DEX, respectively [8]. These *in vitro* HTM spheroid models were also used for studies to evaluate effects on the HTM of various drugs and factors including prostaglandin FP and EP2 agonist [124,125,126,127], autotaxin [128], ATRA [58], benzalkonium chloride (BAC) [129], ROCK inhibitors [130,131,132,133], α2-adrenergic agonist [134] and TGF-β isoforms [135,136,137]. The retinal pigment epithelium (RPE) is a highly organized and polarized single-cell sheet adjacent to the retinal choroid and photoreceptor outer segment (OS), and it functionally plays critical roles in the visual retinoid cycle, phagocytosis of the OS, formation of the outer blood-retinal barrier (oBRB) with Bruch’s membrane and retinal choroid, and transportation of various nutrients [138]. In turn, dysfunctions in the RPE, which can lead to disruption of the oBRB and photoreceptor cell death, are implicated in serious retinal diseases such as age-related macular degeneration (ARMD) and retinitis pigmentosa [139,140]. Therefore, for targeting the RPE, pilot therapies using hiPSCs or human embryonic stem cells (hESCs) have already been started [141,142,143]. To support this issue, various studies using stem cell-based 3D culture of the RPE have been carried out [144,145,146,147,148,149]. In addition, spheroid culture has also been used to replicate RPE-induced angiogenesis [150], proliferative vitreoretinopathy [151] and drug-induced toxicity to the RPE [152].

### 4.3. Orbit

Among various orbital tissues, the lacrimal gland (LG) and orbital fatty tissues have been frequently used for generation of *in vitro* spheroid models.

#### 4.3.1. Lacrimal Gland (LG)

LG dysfunction is one of major causes of dry eye disease and is therefore a possible promising therapeutic target for dry eye disease [153]. For this purpose, 3D organoid cultures have mainly been performed using stem cells [154,155,156,157,158] or digested cells from human LG tissue [159]. Spheroids of LG-related cells including human epithelial cells, endothelial cells and mesenchymal stem cells [160] or rabbit conjunctival epithelium and lacrimal gland cells [104] have also been generated by using a static suspension culture method.

#### 4.3.2. Orbital Fatty Tissue

A major disease of orbital fatty tissue is Graves’ orbitopathy (GO), for which no animal model has been developed. A suitable *in vitro* model is therefore necessary for obtaining a better understanding of the disease etiology and for developing a new therapy. For this purpose, by using a static suspension method, spheroids were generated from human orbital fibroblasts (OFs) obtained from patients without GO (N-OFs) and patients with GO (G-OFs) (Figure 6) [161]. Our group has also generated spheroids from N-OFs and G-OFs spheroids by a hanging drop culture method and we found several new biological differences between N-OFs and G-OFs [5,162,163,164,165,166,167,168]. As for the molecular pathology of GO, autoimmune-based overstimulation of the thyroid-stimulating hormone (TSH) receptor (TSHR) and insulin-like growth factor 1 (IGF1) receptor (IGF1R), both of which are expressed in OFs derived from the monocyte lineage and form a complex by a crosstalk mechanism [169], induces inflammation and differentiation into adipocytes and myofibroblasts as well as hyaluronan production [170,171,172]. 

Deepening of the upper eyelid sulcus (DUES) is known as a notable adverse effect of prostaglandin-derived anti-glaucoma medication with cosmetically annoying problems. Our group first established an *in vitro* spheroid model of DUES using N-OFs and 3T3-L1 cells and we provided valuable information for selection of anti-glaucoma medications to avoid DUES [42,72,73,74,75,162,163,164,165,166,167,168,173,174,175].

## 5. Three-Dimensional Spheroids Obtained from Various Cancerous Cells

As in the case of non-cancerous tissues, 3D culture systems are also being increasingly used in tumor research for establishing essential strategies [176,177]. As shown in Table 1, various methods have been used for generating single cell type-derived spheroids from cancerous cells as in the case of non-cancerous cells, and the spheroids obtained have been mostly used for drug screening to evaluate toxicity and to discover new anti-cancer drugs [176,177]. Among the various methods for generating spheroids from tumor cells, our group used a hanging drop culture method as described above, and we found that 3D configurations of non-cancerous cell-derived spheroids (globe shape, Figure 6) were significantly different from those of cancerous cell-derived spheroids (non-globe shape, Figure 7) [5,7,55,72,109,125]. A non-globe shape appearance was also observed in some tumor cells-derived spheroids generated by using a hanging drop culture method by other research groups [178,179,180,181,182]. Since, as stated above, the hanging drop culture method is the simplest method for generating spheroids that requires only gravity force and buoyant force, the non-spherical shape of spheroids may be caused by deviations from sphericity of the aggregation of cancerous cells. If this speculation is correct, such deviations from sphericity in the tumor cell-derived spheroids could be a key index that reflects some important biological aspects of tumorigenesis. 

### 5.1. Malignant Melanoma (MM)

An MM originating from melanocytes is a very aggressive malignant tumor and occurs in various organs with melanin pigment including the skin, eye choroid, gastrointestinal tract, genitalia, sinuses, and meninges [183]. As recent therapies for MM, targeted inhibitors of BRAFV600E, MEK kinases and immune-based monoclonal antibody drugs have been effectively used for treating patients with MM [184,185,186,187]. However, a relatively large percentage of patients with MM show resistance to these treatments, and the development of an additional therapeutic strategy and the development of a method for control of drug resistance by using a reliable *in vitro* model such as a spheroid model are therefore necessary [188,189,190]. Recently, our group also prepared spheroids originating from 5 different MM cell lines including SK-mel-24, MM418, A375, WM266-4 and SM2-1 by using a hanging drop culture method [7]. Interestingly, the spheroid configurations were deformed and the degree of the deformity was significantly diverse among the 5 MM cell lines. In addition, cellular metabolic functions assessed by using a seahorse bioanalyzer were also different among the MM cell lines. RNA sequencing analysis of two distinct cell lines, WM266-4 and SK-mel-24, in the 3D appearances suggested that KRAS and SOX2 were potential master regulatory genes causing these diverse 3D configurations. Furthermore, knockdown of both factors reduced the morphological deformity and improved functional characteristics. In addition, ML329, a small molecule of a specific inhibitor of microphthalmia-associated transcription factor (MITF) that regulates the expression of various pigmentation-related genes including genes encoding tyrosinase [191] and is critically involved in the pathogenesis of MM [192,193], significantly and diversely modulated the spheroid appearance and cellular metabolic functions of various MM cell lines [54]. Collectively, the findings suggest that spheroid configuration may become a potential key indicator for pathophysiological activities associated with MM. 

### 5.2. Oral Squamous Cell Carcinoma (OSCC)

Oral squamous cell carcinoma (OSCC) is a mucosal malignancy of the lips and oral cavity and is one of main causes of mortality worldwide [194,195]. The 5-year survival rate of patients with OSCC is less than 50% regardless of the stage [196,197,198]. In this research field, 3-D cell cultures have been increasingly used with the aim of establishing physiologically relevant *in vitro* models for studying various aspects of oral cancer [199]. As in the case of MM, deformed appearance of spheroids generated by a hanging drop culture method was also observed in pathologically different squamous cell carcinoma (OSCC) cell lines and one oral adeno squamous carcinoma (OAdSCC) cell line, and the degree of deformity correlated with the efficacy of the CDDP-induced cytotoxicity among the cell lines. Therefore, we suggested that spheroid architectures may be useful indicators for estimating drug sensitivity of OSCC and OAdSCC malignancies [55]. Collectively, the findings suggested that *in vitro* spheroid models may be more sensitive and suitable than *in vitro* conventional 2D culture models for achieving more precise screening, estimating appropriate dosage and estimating efficacies of new candidate antitumor agents [200,201,202,203,204]. 

### 5.3. Cancer-Associated Fibroblasts (CAFs)

Tumor surrounding microenvironments (TSM) have attracted much interest due to their crucial roles in tumorigenesis, tumor progression, and metastasis [205,206,207,208]. In the research field of TSM, it is thought the use of 3D cell culture methods can replicate the tumor microenvironment landscape and are therefore suitable for screening immunomodulatory drugs [209]. Among TSM, specific fibroblasts called “cancer-associated fibroblasts (CAFs)” are known to be critical participants that are mainly responsible for producing various extracellular matrix (ECM) proteins in the TSM [210,211,212,213,214]. Several recent studies have shown that fibroblasts not related to cancer (non-CAFs) may be transformed into CAFs in the TSM [215], and, in fact, recent transcriptome analyses have shown different gene expression profiles in non-CAFs and CAFs [206,216,217,218,219]. However, despite great pathological contributions of CAFs to cancer biology, the only method for identifying CAFs is to use certain markers with low specificities such as α-smooth muscle actin (αSMA) and fibroblast-activating protein-*α* (FAPα) [220]. In our recent study using a hanging drop culture method, we found that, unlike globe-shaped non-CAF spheroids, non-globe-shaped CAF spheroids were generated from four lines of CAFs obtained from the neighborhood of OSCCs and that the deformity and stiffness of the spheroids were different [9]. Therefore, the findings suggest that a spheroid culture method provides a new strategy for characterizing and specifying CAFs. 

## 6. Summary of Current Concepts of the Biological Significance of Hanging Drop Cultures of Single Cell Type-Derived Spheroids and Their Future Prospects

In addition to conventional 2D cell culture, various 3D cell culture techniques provide new possibilities for unidentified biological aspects for advancing biological research. In fact, the use of various 3D culture methods has resulted in the establishment of better strategies for testing drug efficacy, the development of disease models that are closer to their *in vivo* conditions to facilitate a deeper understanding of disease etiology, and a better way to deal with stem cells for future regeneration medicine. However, the benefits and disadvantages of 3D cell cultures, even the simplest single cell type-derived spheroid culture, have not been fully characterized. As an identified disadvantage of spheroids, the possible inability to follow hanging-drop cultures over time should be considered compared to *in vitro* 2D culture models, and proper timing should be therefore adjusted for the most suitable experimental purpose. As shown in this review, our experience in the use of a hanging drop culture method to generate spheroids from various cells in proper timing has shown that the biological natures of spheroids are totally different from those of 2D cultured cells. The benefits of spheroids will lead to further improvements for tissue engineering, modeling of non-cancerous and cancerous diseases, and testing biological functions and drug-induced effects.

## Figures and Tables

**Figure 2 cells-13-01549-f002:**
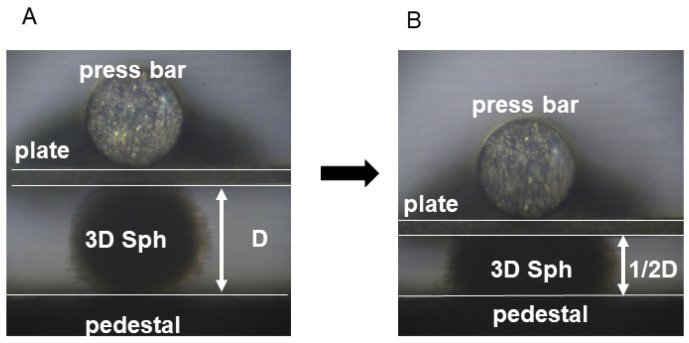
Measurement of stiffness of spheroids. A living spheroid (3D sph) was initially held between a pedestal and a pressing plate ((**A**), D: diameter) and was gradually pressed by a press bar until the initial diameter was halved (D/2) over a period of 20 s (**B**).

**Figure 3 cells-13-01549-f003:**
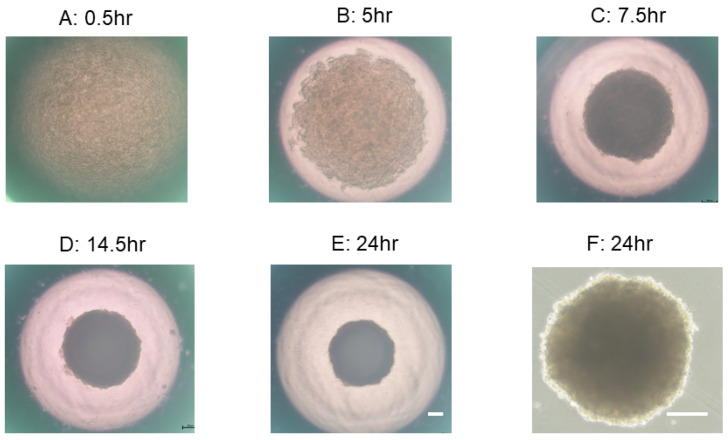
Representative maturation process of H9c2 spheroids. Representative phase contrast (PC) images of a H9c2 spheroid in a well of a hanging drop culture plate during different culture periods at 0.5 h (**A**), 5 h (**B**), 7.5 h (**C**), 14.5 h (**D**), and 24 h (**E**). Panel F shows a representative PC image of a H9c2 spheroid taken out from the culture plate after 24 h culture period (**F**). Scale bar: 100 μm.

**Figure 4 cells-13-01549-f004:**
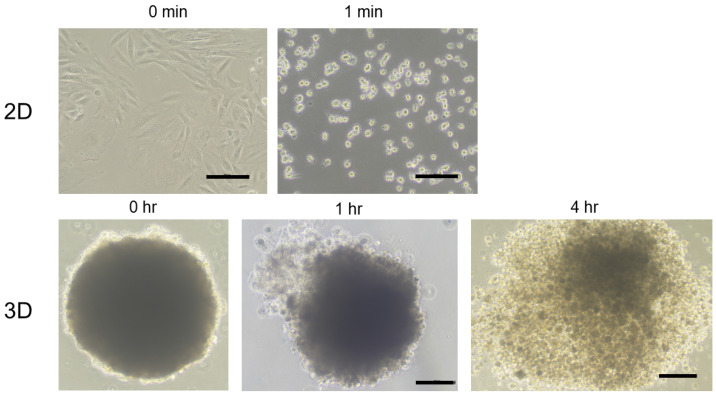
Trypsin digestion of 2D H9c2 cells and 3D H9c2 spheroid. Representative phase contrast images of 2D and 3D cultured H9c2 cells treated by 0.05% trypsin for different time periods are shown. Scale bar: 100 μm.

**Figure 5 cells-13-01549-f005:**
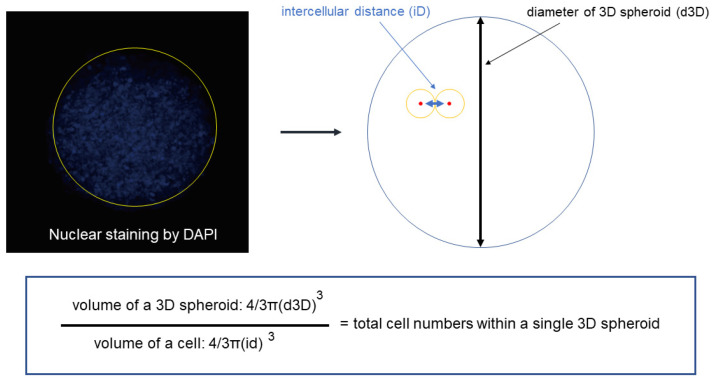
Estimation of total number of cells in a single 3D spheroid.

**Figure 7 cells-13-01549-f007:**
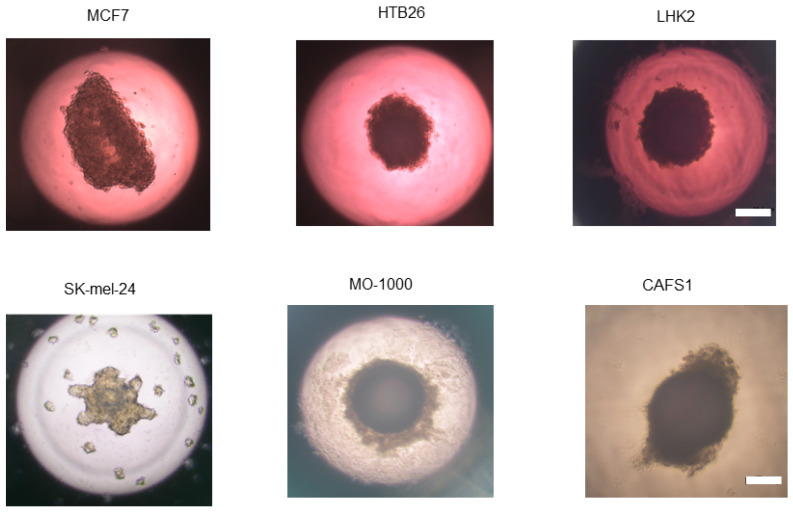
Representative images of 3D spheroid obtained from various non-cancerous cells. MCF7: human breast cancer cell line, HTB26: breast adenocarcinoma cell line, LHK2: lung adenocarcinoma cell line, SK-mel-24: malignant melanoma cell line, MP-1000: oral squamous cell carcinoma cell line, CAFS1: cancer-associated fibroblast cell line. Scale bar: 100 μm.

**Table 1 cells-13-01549-t001:** Various methods for 3D spheroid generation.

	Methods	Place to Generate 3D Spheroid	Effects of Gravity Force	Effects of Buoyant Force	Contribution of Additive Factors
scaffold-assisted					
	hydrogel-assisted culture	on the ground	less	low	hydrogels
	biofilm-assisted culture	on the ground	less	low	biofilms
	particle-assisted culture	on the ground	less	low	particles
	magnet particle-assisted culture	above the ground	moderate	strong	magnetic particles, magnetic force
scaffold non-assisted					
	static suspension culture	on the ground	less	low	none
	floating culture	above the ground	moderate	strong	centrifugal force
	hanging drop culture	under the ground	strong	moderate	none

## Data Availability

The data that support the findings of this study are available from the corresponding author upon reasonable request.

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
