# Peer review of "Application of Single Cell Type-Derived Spheroids Generated by Using a Hanging Drop Culture Technique in Various In Vitro Disease Models: A Narrow Review"

_cells, 2024, doi:10.3390/cells13181549_

Round 1

Reviewer 1 Report

Comments and Suggestions for Authors

Ohguro et al. present a review about the production of spheroids, mainly using the hanging drop technique, and the use of the formed spheroids to investigate adipogenesis, cardiology, ocular pathophysiology and cancerology. The protocol to produce spheroids by the hanging drop technique and to analyze the diameter, morphology, stiffness and metabolism of spheroids is useful detailed for beginners in the field. Interestingly, the responses of spheroids and 2D cell cultures to drug treatment are different. In addition, most cells produce spherical spheroids, except cancerous cells which form non-spherical spheroids.

While the hanging drop technique is extensively described in the manuscript, other techniques to form spheroids are presented too shortly, with only between 1 and 3 sentences per technique.

Especially, the aggregation of cells in the presence of gelatin microparticles (and not nanoparticles as wrongly termed in the section 1.1.3) is unclear, as well as the title “Non-magnetic particle-assisted culture" which is not informative of this approach. The effect of gelatin microparticles to the aggregation of cells should be better addressed.

The “static suspension culture” could discuss Ultra Low Attachment (ULA) plates and/or plates with microcavities.

In the section 1.3 comparing the various techniques of spheroid formation, it is surprising to read that the synthetic scaffolds are unphysiological (line 124: “unphysiologically influenced by some scaffolds”).

The authors use the term “single cell-derived 3D spheroid” throughout the manuscript, including in the title of the manuscript. As spheroids are formed by the aggregation of cells, the term “single cell” is incorrect.

In addition, as spheroids are always in 3D, the term “3D spheroid” can be simplified to just "spheroid".

Figure 2, Line 196: Please use the term "D/2" instead of “1/2D".

Figure 5: the two formulas used to calculate the total number of cells are incorrect. The formulas should be 4/3.π(d3D/2)3 and 4/3.π(id/2)3

Lines 391-392: This sentence is unclear.

Line 449: “the appearance of 3D spheroids”. Do you want to refer to the non-spherical shape of spheroids made by the aggregation of cancerous cells?

Line 452: a mistake is present in the title of the legend: “obtained from various cancerous cells”.

Comments on the Quality of English Language

Minor corrections:

- The manuscript frequently inverts the singular and plural form: "cell cultures" (line 51), "applications" (line 52-53), "spheroid cultures" (line 165), "fibroblasts" (lines 168, 169, 300, 301 twice), "the total number of cells" (lines 220, 225, 228 and 239), "volume" (line 229), "cells" (line 252 twice, lines 254, 255, 299), "spheroids" (lines 272, 297, 446, 452), "were much smaller" (line 290), "studies using different" (line 318), "were observed" (line 327), "our group has" (line 409), "are therefore" (line 467).

- Line 29: "the establishment of"

- Lines 31-34: this sentence has no verb.

- Line 39: "each method has its own advantages"

- Line 58: "there are"

- Line 97: "may induce changes in the cellular biological nature that results in apoptosis"

- Line 115: "density of the cell suspension". Do you mean the concentration of the cell suspension?

- Table 1: the word “less” used several times in the table should be replaced by “low”. "Contribution of unphysiological factors”. The word “hydrogel” is wrongly written. For the magnet particle assisted technique and floating technique, could “above the ground” be more appropriate?

- Line 144: "from the 2D cell culture plate"

- Line 149: "20,000 cells"

- Line 155: "paid to performing"

- Line 164: "was placed to" could be simply changed to "formed".

- Line 166: "a drop"

- Line 184: "a microscope with a camera outside the chamber"

- Line 195: "half of the initial diameter"

- Line 208: "form an immature spheroid"

- Line 218: "100 µm"

- Line 223: “than in the case”

- Line 226: “cross section in a Phalloidin image"

- Line 246: “this system was optimized"

- Line 250: “our group has also”

- Line 280: "suggesting that"

- Line 344: “In vitro" in italics.

- Line 349: add a closing bracket after “isoforms".

- Line 366: “TGF-β2”

- Line 368: "resulting into increased resistance”

- Lines 427 and 428: “Graves” (with a capital “G”)

- Line 450: "tumorigenesis"

- Line 461: "A MM originating from melanocytes is a very aggressive"

- Line 465: "relatively large"

- Line 490: “with the aim"

- Line 497: “drug sensitivity"

- Lines 497-500: does this sentence compare spheroids to 2D cultures?

- Line 515: What is the sign after “fibroblast activation protein”?

Author Response

Dear Editor,

Thank you very much for the constructive comments concerning our manuscript “Application of the single cell-derived 3D spheroid culture technique to various in vitro disease models: A narrow review.”. We carefully checked all of the reviewer’s comments and prepared a revised version of our paper that takes these comments into account. The changes are listed below. In addition, as suggested, English is edited by a native English-speaking scientist and his certificate is attached.

Reviewer 1 comments

Ohguro et al. present a review about the production of spheroids, mainly using the hanging drop technique, and the use of the formed spheroids to investigate adipogenesis, cardiology, ocular pathophysiology and cancerology. The protocol to produce spheroids by the hanging drop technique and to analyze the diameter, morphology, stiffness and metabolism of spheroids is useful detailed for beginners in the field. Interestingly, the responses of spheroids and 2D cell cultures to drug treatment are different. In addition, most cells produce spherical spheroids, except cancerous cells which form non-spherical spheroids.

  1. While the hanging drop technique is extensively described in the manuscript, other techniques to form spheroids are presented too shortly, with only between 1 and 3 sentences per technique. Especially, the aggregation of cells in the presence of gelatin microparticles (and not nanoparticles as wrongly termed in the section 1.1.3) is unclear, as well as the title “Non-magnetic particle-assisted culture" which is not informative of this approach. The effect of gelatin microparticles to the aggregation of cells should be better addressed.

Answer; We sincerely appreciate your excellent comment. As pointed out, non-magnetic particle-assisted culture is changed to gelatin microparticles-assisted culture. As far as I know, almost no study has been available to compare 3D spheroid generation using different methodologies, although most of studies related to 3D spheroid have been performed using single methodology. In addition, many excellent reviews have also already described details of advantages and disadvantages among various methods to generate 3D spheroid. However, I believe almost no review article focusing a hanging drop culture method to generate 3D spheroid using various cell sources has been available, and therefore minimized sections related to other methods. 

  1. The “static suspension culture” could discuss Ultra Low Attachment (ULA) plates and/or plates with microcavities.

Answer; We sincerely appreciate your excellent comment. As suggested, information related to ‘Ultra Low Attachment (ULA) plates and/or plates with microcavities’ is included: ‘In this culture method, cells are concentrated to form a spheroid on culture plates with non-adhesive surfaces such as ultra-low attachment (ULA) plates with U-shaped bottomed wells or surfaces coated with hydrophilic substances such as agar or poly-Hema to minimize cell–substrate adhesion to allow so as cells to remain suspended. Subsequent centrifugation is required to support aggregation of cells and single spheroid formation per well [33]. This culture method has been shown to generate reproducible spheroids with uniform sizes and shapes that are suitable for high-throughput drug screening [34]. To overcome the problem of limited generation of spheroids, a new 96-well plate microcavity arrays-based platform to generate spheroids in the microcavities has recently been developed for drug testing with higher parallelization capacity than that of previous devices [35, 36].’.

  1. In the section 1.3 comparing the various techniques of spheroid formation, it is surprising to read that the synthetic scaffolds are unphysiological (line 124: “unphysiologically influenced by some scaffolds”).

Answer; We sincerely appreciate your excellent comment. As pointed out, I agree that ‘unphysiological factors’ may lead misunderstanding, and therefore, this is changed to ‘additive factors’.

  1. The authors use the term “single cell-derived 3D spheroid” throughout the manuscript, including in the title of the manuscript. As spheroids are formed by the aggregation of cells, the term “single cell” is incorrect. In addition, as spheroids are always in 3D, the term “3D spheroid” can be simplified to just "spheroid".

Answer; We sincerely appreciate your excellent comment. As pointed out, I agree that ‘single cell-derived’ may lead misunderstanding and therefore, this is changed to ‘single cell type-derived’. In addition, 3D spheroids are also changed to just ‘spheroids’.

  1. Figure 2, Line 196: Please use the term "D/2" instead of “1/2D".

Answer; We sincerely appreciate your excellent comment. As pointed out, this is properly corrected.

  1. Figure 5: the two formulas used to calculate the total number of cells are incorrect. The formulas should be 4/3.π(d3D/2)3 and 4/3.π(id/2)3

Answer; We sincerely appreciate your excellent comment. As pointed out, this is properly corrected.

  1. Lines 391-392: This sentence is unclear.

Answer; We sincerely appreciate your excellent comment. As pointed out, this sentence is changed to ‘Among various orbital tissues, the lacrimal gland (LG) and orbital fatty tissues have been frequently used for generation of in vitro spheroid models.’.

  1. Line 449: “the appearance of 3D spheroids”. Do you want to refer to the non-spherical shape of spheroids made by the aggregation of cancerous cells?

Answer; We sincerely appreciate your excellent comment. Yes, I mean “the appearance of 3D spheroids” as “the non-spherical shape of spheroids made by the aggregation of cancerous cells “. Therefore, this sentence is changed to ‘Since, as stated above, the hanging drop culture method is the simplest method for generating spheroids that requires only gravity force and buoyant force, the non-spherical shape of spheroids may be caused by deviations from sphericity of the aggregation of cancerous cells. If this speculation is correct, such deviations from sphericity in the tumor cell-derived spheroids could be a key index that reflects some important biological aspects of tumorigenesis.’.

  1. Line 452: a mistake is present in the title of the legend: “obtained from various cancerous cells”.

Answer; We sincerely appreciate your excellent comment. As pointed out, this is properly corrected.

Comments on the Quality of English Language

Minor corrections:

  1. The manuscript frequently inverts the singular and plural form: "cell cultures" (line 51), "applications" (line 52-53), "spheroid cultures" (line 165), "fibroblasts" (lines 168, 169, 300, 301 twice), "the total number of cells" (lines 220, 225, 228 and 239), "volume" (line 229), "cells" (line 252 twice, lines 254, 255, 299), "spheroids" (lines 272, 297, 446, 452), "were much smaller" (line 290), "studies using different" (line 318), "were observed" (line 327), "our group has" (line 409), "are therefore" (line 467).

Answer; We sincerely appreciate your excellent comment. As pointed out, inverted the singular and plural form are properly corrected.

  1. - Line 29: "the establishment of"

Answer; We sincerely appreciate your excellent comment. As pointed out, this is properly corrected.

  1. - Lines 31-34: this sentence has no verb.

Answer; We sincerely appreciate your excellent comment. As pointed out, this is properly corrected to ‘However, one difficulty in the use of 3D culture is selection of the appropriate 3D cell culture technique for the study purpose among the various techniques ranging from the simplest single cell type-derived spheroid culture to the more sophisticated organoid cultures.’.

  1. - Line 39: "each method has its own advantages"

Answer; We sincerely appreciate your excellent comment. As pointed out, this is properly corrected.

  1. - Line 58: "there are"

Answer; We sincerely appreciate your excellent comment. As pointed out, this is properly corrected.

  1. - Line 97: "may induce changes in the cellular biological nature that results in apoptosis"

Answer; We sincerely appreciate your excellent comment. As pointed out, this is properly corrected.

  1. - Line 115: "density of the cell suspension". Do you mean the concentration of the cell suspension?

Answer; We sincerely appreciate your excellent comment. As pointed out, this is properly corrected to ‘It is possible to control the spheroid size by adjusting the volume of the droplet or the concentration of cells of the suspension.’.

  1. Table 1: the word “less” used several times in the table should be replaced by “low”. "Contribution of unphysiological factors”. The word “hydrogel” is wrongly written. For the magnet particle assisted technique and floating technique, could “above the ground” be more appropriate?

Answer; We sincerely appreciate your excellent comment. As pointed out, there are properly corrected. 

  1. - Line 144: "from the 2D cell culture plate"

Answer; We sincerely appreciate your excellent comment. As pointed out, this is properly corrected.

  1. - Line 149: "20,000 cells"

Answer; We sincerely appreciate your excellent comment. As pointed out, this is properly corrected.

  1. - Line 155: "paid to performing"

Answer; We sincerely appreciate your excellent comment. As pointed out, this is properly corrected.

  1. - Line 164: "was placed to" could be simply changed to "formed".

Answer; We sincerely appreciate your excellent comment. As pointed out, this is properly corrected.

  1. - Line 166: "a drop"

Answer; We sincerely appreciate your excellent comment. As pointed out, this is properly corrected.

  1. - Line 184: "a microscope with a camera outside the chamber"

Answer; We sincerely appreciate your excellent comment. As pointed out, this is properly corrected.

  1. - Line 195: "half of the initial diameter"

Answer; We sincerely appreciate your excellent comment. As pointed out, this is properly corrected.

  1. - Line 208: "form an immature spheroid"

Answer; We sincerely appreciate your excellent comment. As pointed out, this is properly corrected.

  1. - Line 218: "100 µm"

Answer; We sincerely appreciate your excellent comment. As pointed out, this is properly corrected.

  1. - Line 223: “than in the case”

Answer; We sincerely appreciate your excellent comment. As pointed out, this is properly corrected.

  1. - Line 226: “cross section in a Phalloidin image"

Answer; We sincerely appreciate your excellent comment. As pointed out, this is properly corrected.

  1. - Line 246: “this system was optimized"

Answer; We sincerely appreciate your excellent comment. As pointed out, this is properly corrected.

  1. - Line 250: “our group has also”

Answer; We sincerely appreciate your excellent comment. As pointed out, this is properly corrected.

  1. - Line 280: "suggesting that"

Answer; We sincerely appreciate your excellent comment. As pointed out, this is properly corrected.

  1. - Line 344: “In vitro" in italics.

Answer; We sincerely appreciate your excellent comment. As pointed out, this is properly corrected.

  1. - Line 349: add a closing bracket after “isoforms".

Answer; We sincerely appreciate your excellent comment. As pointed out, this is properly corrected.

  1. - Line 366: “TGF-β2”

Answer; We sincerely appreciate your excellent comment. As pointed out, this is properly corrected.

  1. - Line 368: "resulting into increased resistance”

Answer; We sincerely appreciate your excellent comment. As pointed out, this is properly corrected.

  1. - Lines 427 and 428: “Graves” (with a capital “G”)

Answer; We sincerely appreciate your excellent comment. As pointed out, this is properly corrected.

  1. - Line 450: "tumorigenesis"

Answer; We sincerely appreciate your excellent comment. As pointed out, this is properly corrected.

  1. - Line 461: "A MM originating from melanocytes is a very aggressive"

Answer; We sincerely appreciate your excellent comment. As pointed out, this is properly corrected.

  1. - Line 465: "relatively large"

Answer; We sincerely appreciate your excellent comment. As pointed out, this is properly corrected.

  1. - Line 490: “with the aim"

Answer; We sincerely appreciate your excellent comment. As pointed out, this is properly corrected.

  1. - Line 497: “drug sensitivity"

Answer; We sincerely appreciate your excellent comment. As pointed out, this is properly corrected.

  1. - Lines 497-500: does this sentence compare spheroids to 2D cultures?

Answer; We sincerely appreciate your excellent comment. As pointed out, this is comparison with 2D culture model, and therefore, this is changed to ‘Collectively, the findings suggested that in vitro spheroid models may be more sensitive and suitable than in vitro conventional 2D culture models for achieving more precise screening, estimating appropriate dosage and estimating efficacies of new candidate antitumor agents [197-201].’.

  1. - Line 515: What is the sign after “fibroblast activation protein”?

Answer; We sincerely appreciate your excellent comment. As pointed out, this is wrong format and therefore is properly corrected.

Reviewer 2 comments

  1. Did you make these figures by yourself or cite from other papers already published? please add citations if you use figures from other papers.

Answer; We sincerely appreciate your excellent comment. We made all figures by ourselves, but not cited.

  1. please add the section about below paper if you could ‘Arai K, Kitsuka T, Nakayama K. Scaffold-based and scaffold-free cardiac constructs for drug testing. Biofabrication. 2021 Jul 22;13(4). doi: 10.1088/1758-5090/ac1257. PMID: 34233316.’

Answer; We sincerely appreciate your excellent comment. As suggested, this paper (ref# 84) is included in H9c2 spheroid section: ‘For studying the pathophysiology of the heart and for drug screening of the heart, much attention has been paid to in vitro 3D cell culture models using cardiomyocytes [83, 84] for replicating in vivo spatial environments in addition to the standardized two-dimensional (2D) cell culture method.’.

Reviewer 3 comments

The review by Ohguro et al. is an interesting survey of the hanging drop method of 3D culture. It does cite a lot of the group's own work, but that is appropriate in the context.  There are some significant weaknesses in the manuscript:

  1. The title - why not make it explicit that the review is really focused on the hanging drop method?

Answer; We sincerely appreciate your excellent comment. As suggested, title is changed to ‘Application of single cell type-derived spheroids generated by using a hanging drop culture technique to various in vitro disease models: A narrow review.’.

  1. The article has a lot of very, very long and complex sentences that are hard for the reader to follow. These are discussed in more detail in the next section.

Answer; We sincerely appreciate your excellent comment. As pointed as blow and also other reviewers, we do best fix these long and complex sentences in addition to English correction by a native English-speaking scientist (his certification is attached). 

  1. The authors do not seem to justify whether the phenotypic changes that they find to be different between 2D and 3D/hanging-drop are ones that make the 3D a better model for (patho)physiology in vivo. This review should include a thoughtful discussion and review of the advantages and disadvantages of 3D/hanging-drop culture in terms of whether it makes a model that is more representative of in vivo. A related concern is that the authors seem to view sphericity of cultures as a desirable phenotype but do not provide any justification for why that is so. To follow on from that, some cultures are referred to as "deformed". That does not seem to be explained but seems to be due deviations from sphericity. Again, the basis for this terminology is not explained.

Answer; We sincerely appreciate your excellent comment. As pointed out, I agree that the terminology of ‘deformed appearance of tumor spheroid’ is ambiguous. Therefore, as your excellent suggestion to use ‘deviations from sphericity’, corresponding last sentences of Spheroids obtained from various cancerous cells are changed to ‘Since, as stated above, the hanging drop culture method is the simplest method for generating spheroids that requires only gravity force and buoyant force, the non-spherical shape of spheroids may be caused by deviations from sphericity of the aggregation of cancerous cells. If this speculation is correct, such deviations from sphericity in the tumor cell-derived spheroids could be a key index that reflects some important biological aspects of tumorigenesis.’.

  1. The authors conclude that: "As shown in this review, our experience in the use of a hanging drop culture method to generate 3D spheroids from various cells has shown that the biological natures of 3D spheroids are totally different from those of 2D cultured cells and may already be like immature organs." [lines 531-533]. It is not clear what criteria the authors are using to justify the statement about immature organs. This is a critical point that the review must address. This conclusion is also at apparent disagreement with the phrasing in the Abstract [lines 44-45].

Answer; We sincerely appreciate your excellent comment. Since this pointed issue is also pointed out by other reviewer, I realize that this in apparently overstatement and therefore, this statement is deleted in both Abstract and Conclusion.

  1. Many group develop 3D models from single cells using proteinaceous gels such as collagens or Matrigel(R) or Cultrex(R), either with embedding or overlay approaches. This seems to be completely missing from the comparison in section 1.1/Table 1.

Answer; We sincerely appreciate your excellent comment. As suggested, Matrigel(R) or Cultrex(R) are included in the hydrogel section: ‘Hydrogels are crosslinked polymer chains with 3D spatial structures that can contain a large amount of water due to their hydrophilic groups such as -NH2, -COOH, -OH, -CONH2, -CONH, and -SO3H [10]. In practical use, these scaffolds are placed in a well [11], in a microfluidic channel [11] or on a micropillar [12]. An advantage of this culture method is that physical properties, including the size and stiffness, of spheroids can be optimized by selecting biomaterials such as alginate [13], fibrin [14], collagen [15], hyaluronic acid [16] and ECM-based hydrogels, such as Matrigel® [17-19] and Cultrex® [20], and their appropriate concentrations for fabrication of 3D spatial scaffolds [21].‘. As far as I know, almost no study has been available to compare 3D spheroid generation using different methodologies, although most of studies related to 3D spheroid have been performed using single methodology. In addition, many excellent reviews have also already described details of advantages and disadvantages among various methods to generate 3D spheroid. However, I believe almost no review article focusing a hanging drop culture method to generate 3D spheroid using various cell sources has been available, and therefore minimized sections related to other methods.  

  1. Table 1 claims that hanging-drop culture is free from "un-physiological" factors. What about the methylcellulose that they use in the media?

Answer; We sincerely appreciate your excellent comment. I agree that we used methylcellulose is used for spheroid stabilizer, and therefore this information is included.

  1. Figure 1 refers to "Representative immunolabeling" but most of what is shown is dyes such as BODIPY, DAPI, phalloidin and so that is not accurate.

Answer; We sincerely appreciate your excellent comment. As pointed out, "Representative immunolabeling images" is changed to ‘Representative images’.

  1. Overall, the review does not discuss the limitations of imaging of hanging-drop cultures. Presumably the poor quality of most of the images in Fig. 3 reflect the difficulties of imaging the cultures in situ, with the only clear image coming as an end-point after harvesting (Fig. 3F). The review should compare the inability to follow hanging-drop cultures over time with live-cell techniques compared to the ability to do that with various other 3D culture techniques.

Answer; We sincerely appreciate your excellent comment. In terms of ambiguity of Fig. 3, similar data using 3T3-L1 cells (ref # 35, PMID: 33750898) was already published, and therefore, this information is included in 2-3 Representative process of spheroid maturation: ‘As for the maturation process of a spheroid using a hanging drop culture method, our group showed that 3T3-L1 preadipocytes started to coalesce at 1 h after the subjecting cells into wells of the culture plate and formed premature spheroids by 6 h. The formation of spheroids became more apparent with formation of a smaller round-shaped and stiffened mature spheroids from day 1 through day 7 of culture [42]. Similar to 3T3-L1 spheroids, as shown in PC images (Fig. 3), after placing 20,000 H9c2 cardiomyoblasts in 28 μl of culture medium into each well of a spheroid drop culture plate, the cells start to aggregate and then gradually form an immature spheroid within 7.5 hours. At day 1, a matured spheroid had formed.’. In addition, the pointed out issue that the inability to follow hanging-drop cultures over time is extremely important point. Therefore, this information is included in the Conclusion: ‘However, the benefits and disadvantages of 3D cell cultures, even the simplest single cell type-derived spheroid culture, have not been fully characterized. As an identified disadvantage of spheroids, the possible inability to follow hanging-drop cultures over time should be considered compared to in vitro 2D culture models, and proper timing should be therefore adjusted for the most suitable experimental purpose. As shown in this review, our experience in the use of a hanging drop culture method to generate spheroids from various cells in proper timing has shown that the biological natures of spheroids are totally different from those of 2D cultured cells. The benefits of spheroids will lead to further improvements for tissue engineering, modeling of non-cancerous and cancerous diseases, and testing biological functions and drug-induced effects.’.

  1. Interesting results are shown in Fig. 7 on interactions with monocytes, but that is due to a more advanced co-culture system and not the simple model that the authors are emphasizing. This description lacks details of where the GFP-monocytes come from, or whether the media needs to be adjusted to accommodate the immune cells.

Answer; We sincerely appreciate your excellent comment. As pointed out, this is a more advanced co-culture system and not published data. Therefore, this figure and related description is removed because such unpublished data should not be included in review article.

  1. The heading for Fig. 8 says "non-cancerous cells", which is presumably a mistake.

Answer; We sincerely appreciate your excellent comment. As pointed out, this is a careless mistake and therefore, ‘non-cancerous’ is changed to ‘cancerous’.

  1. "fibroblast activating protein-??? (FAP)" is presumably a typo [line 515]; similarly "TGF-??? 2" [line 366]

Answer; We sincerely appreciate your excellent comment. As pointed out, these type errors are fixed.

Comments on the Quality of English Language

Many hard to follow word uses and phrasings, e.g.,

  1. Line 29: "...enable the establish of in vitro models..."

Answer; We sincerely appreciate your excellent comment. As pointed out, this is changed to ‘enable us to establish in vitro model…’

  1. Lines 44-45: "...and are no longer like an immature organ but not merely cells."

Answer; We sincerely appreciate your excellent comment. As pointed out, this may be overstatement and therefore, this phrase is deleted.

  1. Lines 529-531: "However, the essential concept of 3D cell cultures, even the simplest single cell-derived 3D spheroid culture, has not been fully identified."

Answer; We sincerely appreciate your excellent comment. As pointed out, this ‘essential concept’ is not suitable and therefore, this sentence is changed to ‘However, the benefits and disadvantages of 3D cell cultures, even the simplest single cell type-derived spheroid culture, have not been fully characterized.’.

  1. Table 1 has typos: "phsiological", "hyrogogel"

Answer; We sincerely appreciate your excellent comment. As pointed out, these careless typo errors are fixed.

  1. Many very long and complex sentences, e.g.,

Answer; We sincerely appreciate your excellent comment. As pointed out, following sentences are more simply changed as follows: Lines 31-34 is changed to ‘However, one difficulty in the use of 3D culture is selection of the appropriate 3D cell culture technique for the study purpose among the various techniques ranging from the simplest single cell type-derived spheroid culture to the more sophisticated organoid cultures.’, Lines 244-249 is changed to ‘In addition, several past studies have elaborated to apply OCR and ECAR measurements in a 3D culture system using single well assays [44-47]. That system was later optimized to measure OCR and ECAR in specimens of multiple spheroids using 96 multi-well plates [48].’, Lines 275-282 is changed to ‘Recently, our group [6] and another study group [62] have shown spontaneous adipogenesis before adipogenic differentiation in 3D 3T3-L1 spheroids but not in 2D cultured 3T3-L1 cells. In fact, proteomics [51] and RNA sequencing analysis [6, 56] have revealed significant changes in various adipogenesis-related factors, inflammatory cytokines, growth factors, transcription factors, mitochondrial and glycolytic related factors, and other factors during spheroid generation [44].’, Lines 286-293 is changed to ‘In addition, our recent RNA sequencing analysis before and after adipogenic differentiation showed that genes related to adipogenesis and lipid metabolism were altered in 3T3L-spheroids compared to those in 2D cultured 3T3-L1 cells, suggesting that a spheroid environment may be more suitable for adipogenesis of 3T3-L1 cells [56]’, Lines 346-352 is changed to ‘Alternatively, our group established an in vitro subconjunctival fibrosis model using human conjunctival fibroblasts (HconF, Fig. 6) by a hanging drop culture method to study the effects of various drugs including transforming growth factor-b (TGF-b) isoforms [104], fibroblast growth factor-2 (FGF-2) [105], Rho-associated coiled-coil forming kinase (ROCK) inhibitors [106], all-trans retinoic acid (ATRA) [107], prostaglandin EP2 and FP2 agonists [108] and rosiglitazone [106].’.

Reviewer 2 Report

Comments and Suggestions for Authors

Did you make these figures by yourself or cite from other papers already published?

please add citations if you use figures from other papers.

please add the section about below paper if you could 

Arai K, Kitsuka T, Nakayama K. Scaffold-based and scaffold-free cardiac constructs for drug testing. Biofabrication. 2021 Jul 22;13(4). doi: 10.1088/1758-5090/ac1257. PMID: 34233316.

Comments on the Quality of English Language

I can understand 

Author Response

(The authors gave the same response as above.)

Reviewer 3 Report

Comments and Suggestions for Authors

The review by Ohguro et al. is an interesting survey of the hanging drop method of 3D culture. It does cite a lot of the group's own work, but that is appropriate in the context.  There are some significant weaknesses in the manuscript:

1. The title - why not make it explicit that the review is really focused on the hanging drop method?

2. The article has a lot of very, very long and complex sentences that are hard for the reader to follow. These are discussed in more detail in the next section.

3. The authors do not seem to justify whether the phenotypic changes that they find to be different between 2D and 3D/hanging-drop are ones that make the 3D a better model for (patho)physiology in vivo.  This review should include a thoughtful discussion and review of the advantages and disadvantages of 3D/hanging-drop culture in terms of whether it makes a model that is more representative of in vivo. A related concern is that the authors seem to view sphericity of cultures as a desirable phenotype but do not provide any justification for why that is so. To follow on from that, some cultures are referred to as "deformed". That does not seem to be explained but seems to be due deviations from sphericity. Again, the basis for this terminology is not explained.

4. The authors conclude that: "As shown in this review, our experience in the use of a hanging drop culture method to generate 3D spheroids from various cells has shown that the biological natures of 3D spheroids are totally different from those of 2D cultured cells and may already be like immature organs." [lines 531-533].  It is not clear what criteria the authors are using to justify the statement about immature organs. This is a critical point that the review must address. This conclusion is also at apparent disagreement with the phrasing in the Abstract [lines 44-45].

5. Many group develop 3D models from single cells using proteinaceous gels such as collagens or Matrigel(R) or Cultrex(R), either with embedding or overlay approaches. This seems to be completely missing from the comparison in section 1.1/Table 1.

6. Table 1 claims that hanging-drop culture is free from "un-physiological" factors. What about the methylcellulose that they use in the media?

7. Figure 1 refers to "Representative immunolabeling" but most of what is shown is dyes such as BODIPY, DAPI, phalloidin and so that is not accurate.

8. Overall, the review does not discuss the limitations of imaging of hanging-drop cultures. Presumably the poor quality of most of the images in Fig. 3 reflect the difficulties of imaging the cultures in situ, with the only clear image coming as an end-point after harvesting (Fig. 3F). The review should compare the inability to follow hanging-drop cultures over time with live-cell techniques compared to the ability to do that with various other 3D culture techniques.

9. Interesting results are shown in Fig. 7 on interactions with monocytes, but that is due to a more advanced co-culture system and not the simple model that the authors are emphasizing. This description lacks details of where the GFP-monocytes come from, or whether the media needs to be adjusted to accommodate the immune cells.

10. The heading for Fig. 8 says "non-cancerous cells", which is presumably a mistake.

11. "fibroblast activating protein-??? (FAP)" is presumably a typo [line 515]; similarly "TGF-??? 2" [line 366]

Comments on the Quality of English Language

1. Many hard to follow word uses and phrasings, e.g.,

Line 29: "...enable the establish of in vitro models..."

Lines 44-45: "...and are no longer like an immature organ but not merely cells."

Lines 529-531: "However, the essential concept of 3D cell cultures, even the simplest single cell-derived 3D spheroid culture, has not been fully identified."

2. Table 1 has typos: "phsiological", "hyrogogel"

3. Many very long and complex sentences, e.g.,

Lines 31-34

Lines 244-249

Lines 275-282

Lines 286-293

Lines 346-352

Author Response

(The authors gave the same response as above.)

Round 2

Reviewer 3 Report

Comments and Suggestions for Authors

The revised manuscript has been significantly improved in response to the previous critiques.

The major outstanding problems are in Figures 7-8. The Figure shown at lines 424-425 is joined by a caption (lines 426-429) that does not match it. In fact, that Figure is stated as deleted in the authors' response. The Figure shown at lines 452-453 is accompanied by an old caption with a significant typo (lines 454-459). The revised caption for that is the one included in the wrong place at lines 426-429.

Comments on the Quality of English Language

Some problems still with how a few characters are formatted. e.g., line 523:

"fibroblast-activating protein-??? (FAP ???)"

Author Response

Dear Editor,

Thank you very much for the constructive comments concerning our manuscript “Application of the single cell type-derived 3D spheroid culture technique to various in vitro disease models: A narrow review.”. We carefully checked all of the reviewer’s comments and prepared a revised version of our paper that takes these comments into account. The changes are listed below.

Reviewer 3 comments

The revised manuscript has been significantly improved in response to the previous critiques.

  1. The major outstanding problems are in Figures 7-8. The Figure shown at lines 424-425 is joined by a caption (lines 426-429) that does not match it. In fact, that Figure is stated as deleted in the authors' response. The Figure shown at lines 452-453 is accompanied by an old caption with a significant typo (lines 454-459). The revised caption for that is the one included in the wrong place at lines 426-429.

Answer; We sincerely appreciate your excellent comment. As pointed out, this careless mistake to remove Fig. 7 and its wrong legend is corrected, and Fig 8 is changed to Fig. 7.

  1. Comments on the Quality of English Language: Some problems still with how a few characters are formatted. e.g., line 523: "fibroblast-activating protein-??? (FAP ???)"

Answer; We sincerely appreciate your excellent comment. As pointed out, this careless wrong format is corrected: ‘fibroblast-activating protein-a (FAPa)’. In addition, I carefully checked whole manuscript to correct other similar t
